# Autonomous underwater adhesion driven by water-induced interfacial rearrangement

Le Yao[1], Chengjiang Lin[2], Xiaozheng Duan [2] ✉, Xiaoqing Ming[1], Zhixuan Chen[1], He Zhu [1], Shiping Zhu[1] & Qi Zhang [1] ✉

Underwater adhesives receive extensive attention due to their wide applications in marine explorations and various related industries. However, current adhesives still suffer from excessive water absorption and lack of spontaneity. Herein, we report an autonomous underwater adhesive based on poly(2-hydroxyethyl methacrylate-*co*-benzyl methacrylate) amphiphilic polymeric matrix swollen by hydrophobic imidazolium ionic liquid. The as-prepared adhesive is tough and flexible, showing little to none instantaneous underwater adhesion onto the PET substrate, whereas its adhesion energy on the substrate can grow more than 5 times to $458\,\text{J}\cdot\text{m}^{-2}$ after 24 hours. More importantly, this process is entirely spontaneous, without any external pressing force. Our comprehensive studies based on experimental characterizations and molecular dynamic simulations confirm that such autonomous adhesion process is driven by water-induced rearrangement of the functional groups. It is believed that such material can provide insights into the development of next-generation smart adhesives.

Recent years have witnessed the rapid progress of underwater adhesives, due to their increasing demand in ocean exploration, as well as promising applications in implantable healthcare monitors, water-based energy devices, underwater sensors and actuators[1–4]. As compared with the commonly used adhesives for dry conditions, developing underwater adhesive is generically more challenging, not only because the hydration layer between adhesive and substrate would severely hinder the interfacial bonding, but also the infiltrating molecules can solvate the adhesive material and cause swelling, which can further result in the adhesion failure[5,6]. On the other hand, the curing treatment for the glue-type adhesive[7–9] and the preload dependence[10] for the tape-type adhesives[11,12] bring in difficulty for adhesive operation in certain complex, harsh and changeable underwater application scenarios. Toward this issue, an ideal underwater adhesive material that could spontaneously anchor and form strong bonding with the substrate is indeed appealing and under high demand. Unfortunately, such autonomous underwater adhesive products are not yet available.

From the perspective of conventional underwater adhesion design, water molecules were usually treated as an unfavorable factor to be eliminated[13,14]. However, to achieve autonomous underwater adhesion (AUA) with increasing adhesion energy during water immersion yet without manual intervention (external force, light or thermal treatment), we would like to take advantage of the ubiquitous water as a beneficial factor for the growth of the adhesive interface. Thus, we hypothesize that a complete growing process of AUA requires the following steps: 1) proper amount of water absorption to assimilate the hydration layer, meanwhile, to activate the adhesive surface; 2) rearrangement of the functional groups at the interface; 3) establishment of strong yet non-erodible interfacial interaction to perform a long-term adhesion (Fig. 1). Significantly, the rearrangement step plays a crucial role in the spontaneity of AUA, where sophisticated amphiphilicity design of the material is needed. Based on the above hypothesis, we propose to introduce amphiphilic ionogel[15] as the candidate material for AUA. In an amphiphilic gel system, the total hydrophobicity and the covalent crosslinks could maintain the bulk

[1]School of Science and Engineering, The Chinese University of Hong Kong, Shenzhen, Shenzhen, Guangdong 518172, P.R. China. [2]State Key Laboratory of Polymer Physics and Chemistry, Changchun Institute of Applied Chemistry, Chinese Academy of Sciences, Changchun, Jilin 130022, P. R. China. ✉ e-mail: xzduan@ciac.ac.cn; qizhang@cuhk.edu.cn

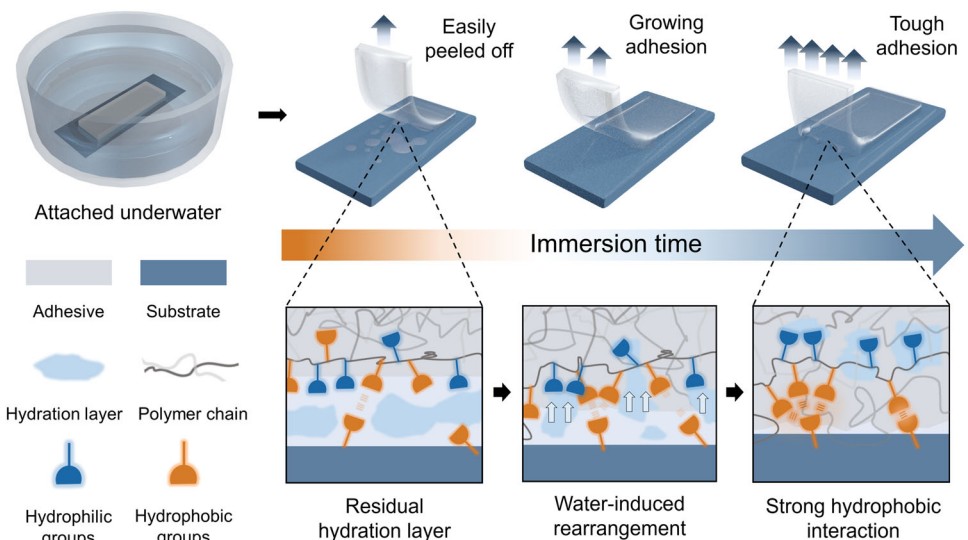

**Fig. 1 | Autonomous underwater adhesion (AUA).** A schematic illustration of a the proposed mechanism of AUA driven by water-induced interfacial rearrangement, involving the expected macroscopic performance and the intrinsic evolution during the water immersion.

integrity and avoid over-swelling, while the hydrophilic component is employed as the regulator of water responsiveness[16]. During water immersion, certain hydrophobic groups (e.g., phenyl) would be freed from the pristine gel system and aggregate[17], which could probably result in the reformation of dynamic interaction at a new site. Moreover, as the solvent of the gel matrix, ionic liquid (IL) could provide free volume and plenty of interactions for the rearrangement[18,19], contributing to the chain mobility and the bulk toughness of the AUA material.

Herein, a unique AUA tape is designed and prepared by incorporating an amphiphilic copolymer matrix into a hydrophobic IL. The as-prepared ionogel tape is almost nonadhesive, but strong adhesion is gradually built during water immersion without any additional loading. Comprehensive studies through both experimental characterizations and molecular dynamic (MD) simulations have been conducted to gain fundamental understanding on this unique adhesion process. Our work provides a new strategy for the development of smart underwater adhesives, which can also largely broaden the application areas of gel materials.

## Results
### Material design and preparation
In previous study, poly(benzyl methacrylate) (PBzMA) and certain imidazolium ILs could form homogeneous phases only in the presence of certain amount of hydrophilic component, i.e., bis(trifluorosufonylmethane imide) lithium salt (LiTFSI)[20], due to the equilibrium stabilized by polymer solvation and interpolymer interactions[21]. When invaded by external water molecules, the unstable ternary system is prone to rearrange and the hydrophobic polymer segments would aggregate and undergo larger-scale phase separation[17,22]. However, such a strongly phase-separated material is too rigid and brittle to be applied as adhesive, and the fast diffusion of LiTFSI in aqueous environment is also undesirable for underwater scenes. Toward this issue, we employed the hydrophilic component as a monomer, covalently bonded to PBzMA polymer chains.

As a typical hydrophilic polymer, poly(2-hydroxyethyl methacrylate) (PHEMA) is not compatible with most hydrophobic ILs, such as 1-ethyl-3-methyl imidazolium bis(trifluorosufonylmethane imide) (EMITFSI)[23]. Surprisingly, the random copolymers of BzMA and HEMA have good miscibility with EMITFSI, indicating a subtle balance of interactions between the polymer and IL (Supplementary Fig. 1). We therefore hypothesize that the breakdown of such balance once

introduced with a poor solvent (water), would bring in possible interfacial interaction rearrangement, which could further offer great potential and opportunity to achieve AUA.

Our AUA materials were prepared through in situ photoinitiated[24–26] copolymerization of BzMA and HEMA in the presence of the ionic liquid EMITFSI, using divinylbenzene (DVB) and 1-hydroxycyclohexyl phenyl ketone (PI-184) as the cross-linker and photoinitiator, respectively (Fig. 2a). The monomer mass fraction is defined by $f = m_{HEMA} / (m_{BzMA} + m_{HEMA})$, the IL content is defined by $I = [m_{IL} / (m_{monomer} + m_{IL})] \times 100\%$, and the crosslinker content is defined by $X = (m_{crosslinker} / m_{monomer}) \times 100\%$. All the samples were prepared following the recipes in Supplementary Table 1. Without specified otherwise, AUA tapes at $f = 0.5$, $I = 40\%$, and $X = 0.15\%$ were chosen throughout the test.

### AUA performance
Before immersing into water, the as-prepared material presents as a tough, flexible and non-adhesive tape and can withstand various mechanical treatments, such as twisting, folding and stabbing (Supplementary Fig. 2). Due to the weak interfacial interaction between the tape and the substrate, the bulk material is barely involved in energy dissipation during peeling, while an obvious dissipation zone could be observed after 24-hour immersion (Fig. 2b). To quantify this behavior, typical 90-degree peeling tests were firstly applied on polyethylene terephthalate (PET) substrates. During the test, all samples presented interfacial failures, breaking neither the adhesives nor the substrates, and no residual was observed on the substrate after peeling (Supplementary Fig. 3). Corresponding curves of force/width versus displacement of the AUA tapes at different immersion time ($t = 0$, 30 min and 24 h) are gathered in Fig. 2c, which perfectly matched our hypothesis. Moreover, this process has no observable change in the size and shape, and the tapes remain excellent transparency (>96%) after 24-hour immersion (Supplementary Fig. 4). For comparison, a reference sample was gently loaded on a dry PET substrate and then sealed for 24 hours, which presented a low adhesion energy ($\Gamma$) during peeling (Supplementary Fig. 5). Evidently, the adhesion performance evolution was triggered by water absorption rather than the simple prolongation of contact time.

To further confirm the spontaneity, $\Gamma$ at different preload strengths (from 1 kPa to 100 kPa) was measured. The result shows no significant correlation between $\Gamma$ and preload strength and no further loading required during the immersion process, indicating that the

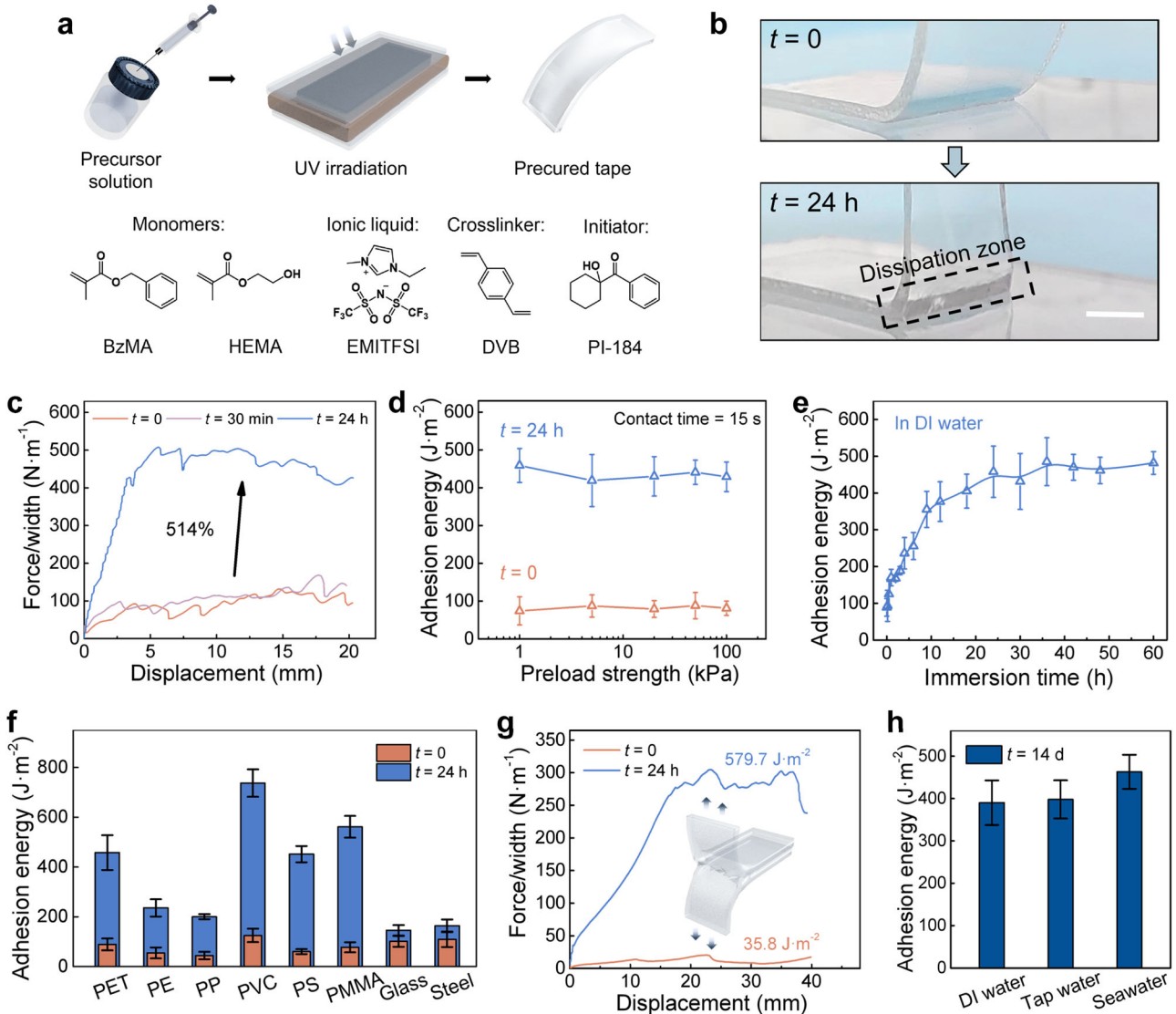

**Fig. 2 | Adhesion performance. a** A schematic illustration of the preparation process of AUA tape, and the molecular structures of the monomers, ionic liquid, crosslinker and initiator in the precursor solution. **b** Photographs comparing $t = 0$ and $t = 24$ h of the AUA during peeling (scale bar = 5 mm). **c** Force-displacement curve recorded in 90-degree peeling test comparing $t = 0$, $t = 30$ min and $t = 24$ h. **d** Adhesion energy ($\Gamma$) of AUA at different preload strengths (1-100 kPa) with a contact time of 15 s, comparing $t = 0$ and $t = 24$ h. **e** $\Gamma$ of AUA versus immersion time. **f** $\Gamma$ comparing the AUA performance on different substrates. **g** Force-displacement curve recorded in 180-degree peeling test comparing $t = 0$ and $t = 24$ h, showing the self-fusing performance of AUA. **h** $\Gamma$ of AUA after 14-day immersion in deionized (DI) water, tap water and artificial seawater, respectively. AUA tapes at $f = 0.5$, $I = 40\%$, and $X = 0.15\%$ were chosen throughout the test; values represent the mean and standard deviation, where $n = 3$.

external pressing force does not play an important role in the AUA process (Fig. 2d). To investigate the time dependence of AUA, the correlation curve of $\Gamma$ and $t$ was plotted (Fig. 2e). The first 12 hours of immersion gave rise to $\Gamma$ rapidly from 89.1 J·m$^{-2}$ to 376.9 J·m$^{-2}$. After 24 hours, $\Gamma$ reached 458.0 J·m$^{-2}$, which is 514% of the $t = 0$ state, then the increase slowed down. This result clearly presents the strong time dependency of $\Gamma$ during the underwater adhesion formation process.

Meanwhile, the adhesion performance of AUA tapes was evaluated on various substrates, including polyethylene (PE), polypropylene (PP), polyvinyl chloride (PVC), polystyrene (PS), polymethyl methacrylate (PMMA), glass and steel (Fig. 2f). On all the commonly used plastic substrates, $\Gamma$ at $t = 24$ h reached 4–6 folds larger than at $t = 0$. Whereas for other substrates, such as glass and steel, $\Gamma$ showed less growth after 24-hour immersion. Obviously, the AUA behavior is more significant on plastic substrates, which indicates that the hydrophobic groups probably play a dominant role in the formation of

strong adhesion. In addition, a 180-degree peeling test was conducted by applying AUA tape itself as the substrate. The result displays good underwater self-fusing performance of AUA tapes (Fig. 2g).

To test the long-term durability, we immersed the AUA tapes with one side attached to the PET substrate in deionized (DI) water, tap water and artificial seawater for 14 days. As shown in Fig. 2h, all the samples can retain high adhesion energy for a long period underwater, providing great availability for various types of underwater applications. Besides, the AUA tapes have good reusability at a short immersion time. Even after 100 misplacements and peelings, the AUA tapes were still able to grow to the same level of $\Gamma$ (465.2 J·m$^{-2}$) after a 24-hour immersion (Supplementary Fig. 6a, b). Moreover, the AUA tapes still have certain repeatability even after strong adhesion has been built on the substrate. For example, after one immerse-peel-dry cycle, 70% of $\Gamma$ could be reserved, as compared with the first growth (Supplementary Fig. 6c, d).

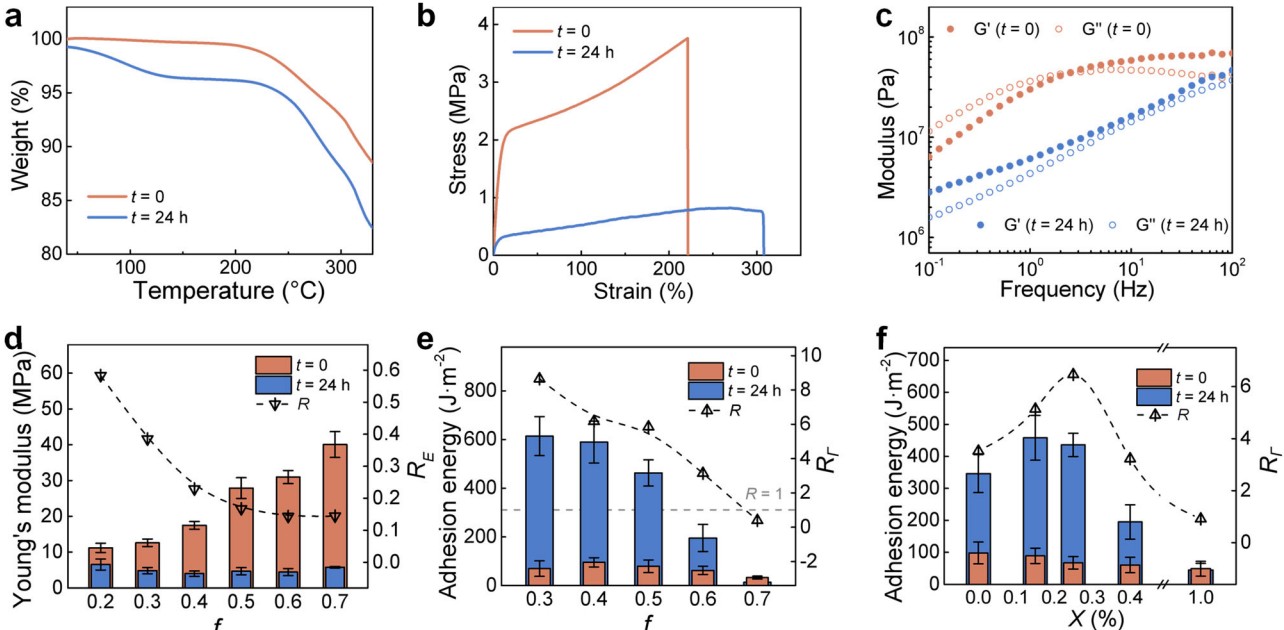

**Fig. 3 | Bulk properties and composition effect. a** TGA curves, **b** typical tensile stress-strain curves and **c** the frequency-sweep dynamic mechanical performance curves with a strain of 0.5% comparing $t = 0$ and $t = 24$ h. **d** Young's modulus and **e** $\Gamma$ on PET substrates comparing $t = 0$ and $t = 24$ h at different monomer fraction ($f$). **f** $\Gamma$ on PET substrates comparing $t = 0$ and $t = 24$ h at different crosslinker content ($X$). The $R_E$ and $R_\Gamma$ are defined as the ratio in Young's modulus and adhesion energy of $t = 24$ h to $t = 0$. Values represent the mean and standard deviation, where $n = 3$.

## Properties of the bulk material

After 24-hour immersion, the water uptake of AUA tape was around 4 wt% as measured by thermogravimetric analysis (TGA) (Fig. 3a). To study how such a small amount of water would affect the bulk material, differential scanning calorimetry (DSC) analysis was employed. Significant decrease in glass transition temperature ($T_g$) after 24-hour immersion from 6.2 °C to -13.0 °C was observed at a heating rate of 10 °C · min⁻¹ (Supplementary Fig. 7). The difference in $T_g$ reveals the water-induced plasticizing effect during the immersion process[27], which could possibly enhance the mechanical interlock between the adhesive and substrate. Accordingly, the mechanical property of the bulk material is altered. From the typical tensile stress-strain curve and loading-unloading curve, AUA tapes at $t = 0$ exhibit higher Young's modulus, tensile strength, bulk toughness, dissipated energy and lower breaking elongation than that after water immersion for 24 h (Fig. 3b and Supplementary Fig. 8).

As for the results of the dynamic mechanical analysis (DMA), the storage modulus ($G'$) at $t = 0$ is larger than at $t = 24$ h over a frequency range of 0.1 Hz to 100 Hz, which highly agrees with the static mechanical performance. At $t = 0$, AUA tape showed a typical glass transition (dissipating) behavior[28] ($G'' > G'$) at low frequency, suggesting that the large amount of sacrifice/dynamic bonds within the network was devoted to energy dissipation. At higher frequencies (> 5 Hz), $G''$ declined since the dissipation had reached a limit and the bulk material started to turn glassy. At $t = 24$ h, $G'$ is larger than $G''$ and both of them increase with frequency, which stands for a typical rubbery state (Fig. 3c). This result reveals that water would affect the dynamic networks maintained by the interaction between functional groups.

Additionally, we analysed the relaxation curves of AUA tape at a small strain, which shows an approximate value (0.02 MPa) between $t = 0$ and $t = 24$ h after a long relaxation time (Supplementary Fig. 9). Since the stress of such a polymer gel system can be simply considered as a sum of a primary network and a dynamic network[29], this result proves that water molecules would disintegrate the dynamic network consisting of sacrificial bonds within the polymer gel during the rearrangement[30,31].

## Composition effect on AUA

To confirm the contribution of the hydrophilic/hydrophobic component within the polymer matrix, we compared the Young's modulus and adhesion energy at different $f$. For $f$ ranging from 0.2 to 0.7, all the AUA samples were softened after 24-hour immersion, with the modulus ratio $R_E$ ($R_E = E_{24h}/E_0$) decreased with $f$ (Fig. 3d). With the increase of $f$, in other words, more hydrophilic components involved, $\Gamma$ decreased significantly both at $t = 0$ and $t = 24$ h (Fig. 3e). The adhesion energy ratio $R_\Gamma$ ($R_\Gamma = \Gamma_{24h}/\Gamma_0$) dropped to <1 when $f$ exceeded 0.7 where the AUA behavior vanished. For the sample with $f$ lower than 0.3, phase separation was observed during the early stage of immersion, which would further cause a failure in removing the hydration layer. These results imply that HEMA plays a crucial role in maintaining the equilibrium of this ternary gel system following the principle below: With a higher $f$, the AUA system is inactive and less water-sensitive while a lower $f$ will make the disruption of this equilibrium by water easier and more drastic.

In the AUA tape, the crosslinker provides chemical crosslinks to the bulk gel, preventing it from larger-scale phase separation during the immersion (Supplementary Fig. 10). Owing to the covalently crosslinked network, there will be only a tiny amount of ionic liquid loss (2.85 wt% after 7 days), which is negligible compared to the total ionic liquid content (40 wt%) (Supplementary Fig. 11). At $X = 0.15\%$ and 0.25%, the samples performed excellent autonomous adhesion. As $X$ increased to 0.4%, the $\Gamma$ gap between $t = 0$ and $t = 24$ h shrunk significantly. Further increasing $X$ to 1.0% could lead to ineffective development of autonomous adhesion (Fig. 3f). The high density of crosslinks would confine the chain movement and aggregation within the gel matrix[32], indicating that the mobility of the polymer chain contributes to the AUA performance. Besides, we also studied the correlation between ionic liquid content and AUA performance. The samples at $I = 40\%$ showed the greatest $R_\Gamma$ (Supplementary Fig. 12) since they were too fragile to form strong adhesion at higher $I$ and too rigid for water to penetrate at lower $I$ which determines the fluidity and stiffness of the gel system[33]. At $I = 0$, the material will turn into a brittle plastic without any adhesion (Supplementary Fig. 13a). Obviously,

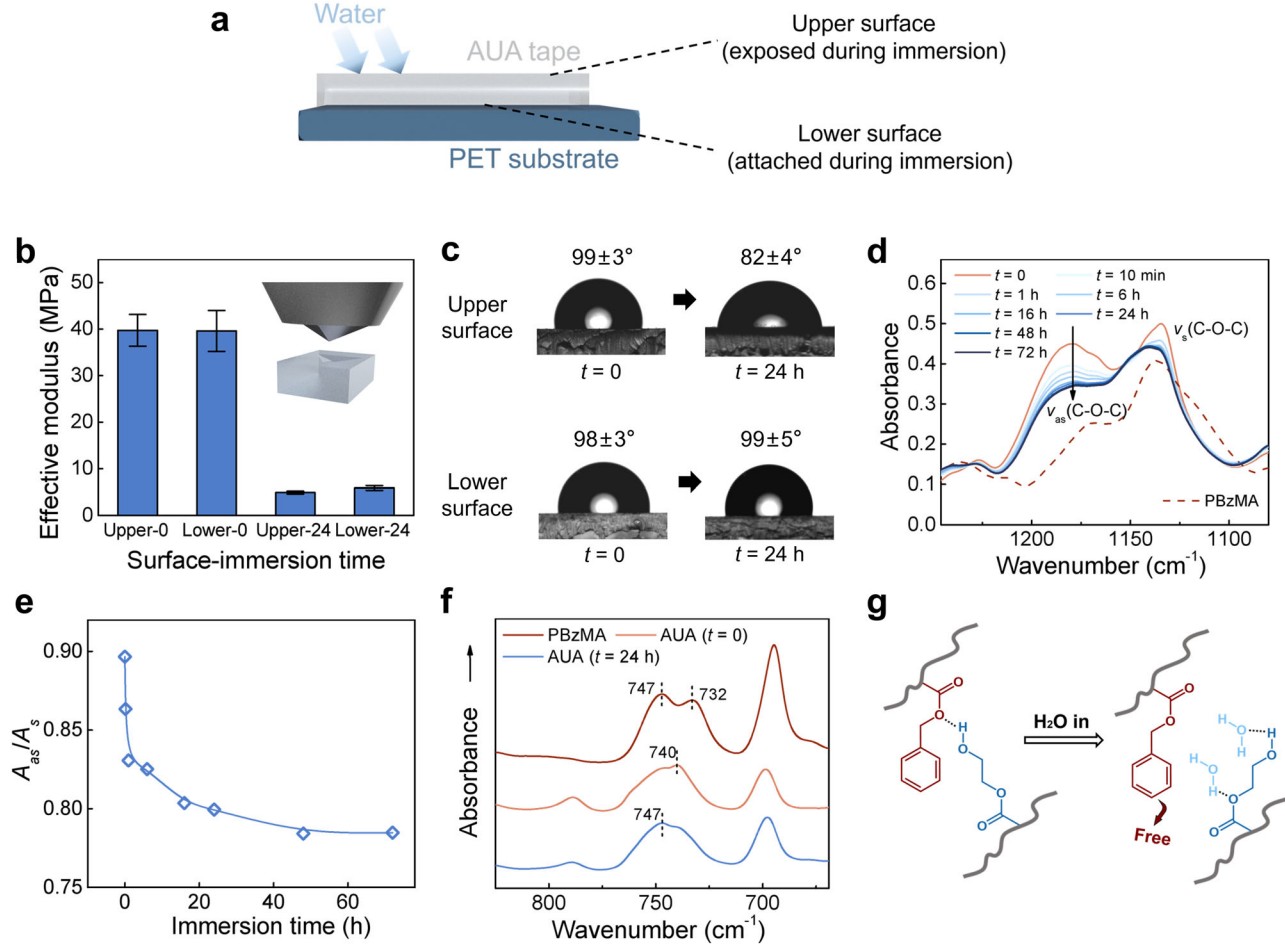

**Fig. 4 | Water-induced surface evolution. a** An illustration of the upper (exposed) surface and the lower (attached) surface of AUA during immersion. **b** Effective modulus ($E_{eff}$) of the surfaces computed from indentation tests. Values represent the mean and standard deviation, where $n = 3$. **c** Evolution of water contact angle on AUA. **d** ATR-FTIR spectra showing the stretching region of C-O-C at different immersion time and **e** the corresponding peak ratio ($A_{as}/A_s$ is defined as the ratio in absorbance of the asymmetric stretching vibration to the symmetric stretching vibration) revolution. **f** ATR-FTIR spectra showing the shift of monosubstituted benzene peak of AUA, compared with the pure BzMA. **g** An illustration of the rearrangement of the ester, benzene and hydroxyl groups when introduced with water. (AUA tapes at $f = 0.5$, $I = 40\%$, and $X = 0.15\%$ were chosen throughout the test.).

EMIFTSI not only provides a subtle dissolution equilibrium and inter-ionic interactions to the AUA process, but also plays an important role in regulating the bulk mechanical properties (Supplementary Fig. 13b).

Drawing from the preceding discussion, we contend that this strategy exhibits the capacity for broad generalization and applic-ability across diverse material systems. Any material system composed of an amphiphilic polymer matrix and an ionic liquid could be AUA, provided parameters *f*, *X* and *I* are appropriately tuned to align with the specific characteristics. Here, we proposed a universal approach to fabricating AUA with different monomers and ionic liquids. Ethyl methacrylate (EMA), dimethylacrylamide (DMAA) and 1-butyl-3-methyl imidazolium hexafluorophosphate (BMIPF₆) were applied to replace the hydrophobic monomer, hydrophilic monomer and ionic liquid in our recipe (Supplementary Table 2), separately, and the result shows considerable AUA performance on PET substrate as well (Supple-mentary Fig. 14).

**Study on the surface evolution of AUA**
Intriguingly, we found that the surface of AUA material exposed to water was slippery rather than adhesive implying that the two surfaces probably underwent distinct evolutions during immersion. This divergence warranted further investigation as it promised valuable insights into the underlying mechanisms driving such anomalous AUA phenomenon. Here, we marked the surface exposed to water during immersion as 'upper', and the surface attached to the substrate as 'lower' (Fig. 4a). The mechanical property evolution of the AUA sur-faces was investigated by an indentation test, and the effective moduli ($E_{eff}$) in MPa were computed following Oliver-Pharr method[34]:

$$E_{eff} = \frac{\sqrt{\pi}}{2} \frac{S}{\beta\sqrt{A}} \tag{1}$$

where $S$ is the slope of the linear interval of the unloading force curve (Supplementary Fig. 15) in mN/μm, $A$ is the contact area in mm² and $\beta$ is a dimensionless parameter of the indenter. Both upper and lower surfaces showed a decline in $E_{eff}$ from 40 MPa to around 5 MPa (Fig. 4b), which basically agrees with the tensile test results, indicating that both surfaces of AUA were effectively plasticized by water.

To study the hydrophobicity evolution of the AUA surface during immersion, contact angles of water on both sides of AUA tapes were measured. Before immersion, AUA tapes were highly hydrophobic, with an average contact angle of almost 100°. After 24-hour immer-sion, the contact angle of the upper surface decreased to 82°, while the lower one retained its original hydrophobicity (Fig. 4c). Besides, the upper surface is fully nonadhesive, indicating that the autonomous adhesion is only formed at specific interfaces. Similar result was obtained when replacing the substrate with glass (Supplementary Fig. 16). As the reference of hydrophobicity, we measured the contact

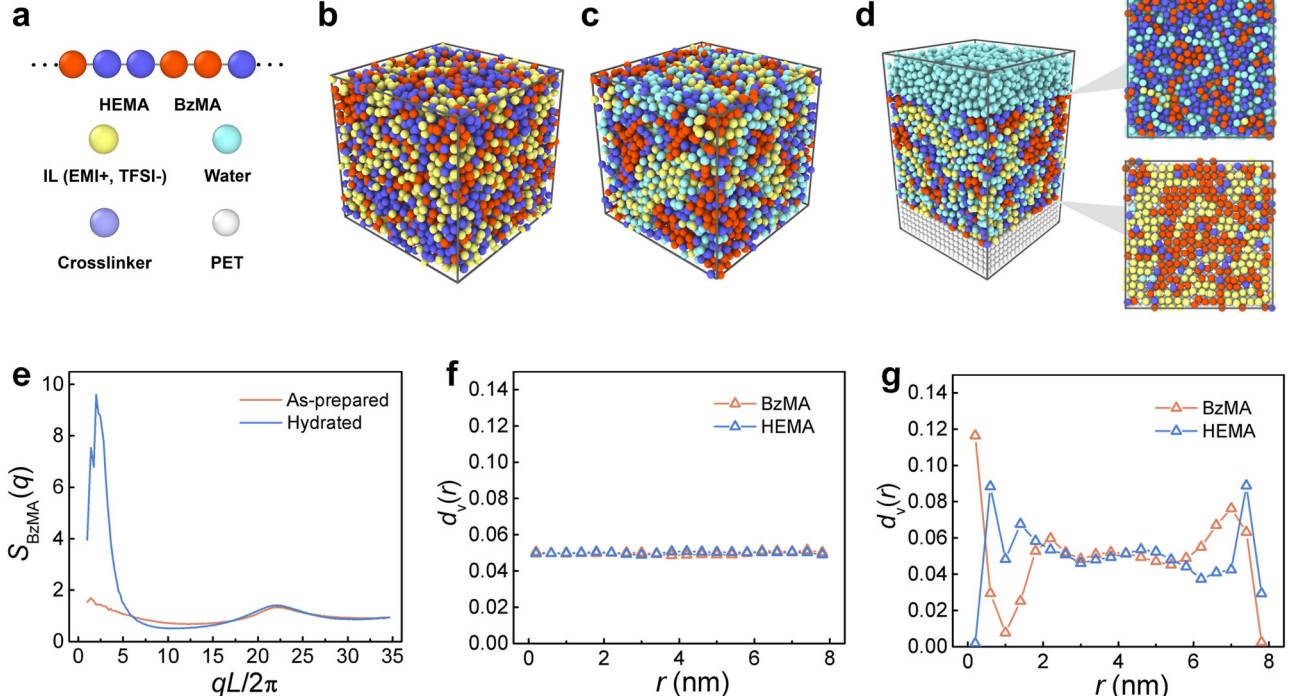

**Fig. 5 | Molecular dynamics (MD) simulation. a** Schematics of a coarse-grained random copolymer composed of BzMA and HEMA, a cation or an anion of IL, a water molecule, a crosslinker, and a PET bead of the hydrophobic substrate. Simulation snapshots of bulk polymer/IL complex at **b** as-prepared, **c** hydrated states and **d** hydrated polymer/IL complex between a water phase and a hydrophobic substrate. The insets show the water-adhesive interface and substrate-adhesive interface of the polymer/IL complex. **e** Static structure factor of BzMA ($S_{BzMA}(q)$) in polymer/IL complex at as-prepared and hydrated states. Vertical distributions $d_v(r)$ of BzMA/HEMA in polymer/IL complexes **f** at the as-prepared state and **g** between a water phase and a hydrophobic substrate where $r = 8$ nm indicates the interface between the complex and water phase and $r = 0$ indicates the interface between the complex and the hydrophobic substrate.

angle of water on all the tested substrates. The water contact angle on the plastic substrates was between 80° to 105°, which is close to the contact angle of AUA, while the glass and steel substrates are more hydrophilic (<55°) (Supplementary Fig. 17). This result could be a rational explanation for why AUA is more significant on plastic substrates. Moreover, such obvious contrast between the two water-plasticized surfaces points to a directional motion of the functional groups during the interfacial adhesion development.

To investigate how the functional groups and interactions rearrange on the AUA surfaces, attenuated total reflectance-Fourier transform infrared (ATR-FTIR) was performed. For pure PBzMA, the characteristic peaks at 1168 cm⁻¹ and 1135 cm⁻¹ were assigned to the asymmetric and symmetric stretching vibration of C-O-C, respectively[35]. The PHEMA and PBzMA-co-HEMA exhibited a single peak of $\nu$(C-O-C) since more C-O-C between the polymer chains were hydrogen bonded[36]. However, for the as-prepared AUA, the double peak of $\nu$(C-O-C) was observed and the $\nu_{as}$(C-O-C) blue shifted to 1179 cm⁻¹, indicating an unstable and highly active system (Supplementary Fig. 18). During water immersion, this intense peak was attenuated, which was gradually fitting the peak profile of pure PBzMA (Fig. 4d). The peak ratio of $\nu_{as}$(C-O-C) to $\nu_s$(C-O-C) decreased from 0.90 to 0.78, and this variation of peak shape reached equilibrium after 48 hours in accordance with the increase of adhesion (Fig. 4e). Furthermore, the monosubstituted peak of benzene for AUA was at a lower wavenumber (740 cm⁻¹) compared to pure PBzMA, and it shifted back to 747 cm⁻¹ after 24-hour immersion (Fig. 4f). This result implies that the BzMA units would tend to separate from the AUA system when the subtle balance was broken down by water molecules (Fig. 4g), possibly leading to an autonomous adhesion when such rearrangement is carried out on a hydrophobic interface (e.g., PET). Interestingly, the competition between the hydrophobicity units (BzMA) and hydrophilic units (HEMA) did not evolve into observable phase

separation. The small-angle X-ray scattering (SAXS) profiles confirmed this statement that neither the pristine sample nor the 24-hour-immersed sample showed any scattering peak (Supplementary Fig. 19). This might be attributed to the randomness of the free radical polymerization and the conformal constrain of the crosslinked polymer chain. Thus, we attempted to use computational methods to elucidate the more microscopic process in the following chapter.

**Molecular dynamics (MD) simulation**

To fundamentally illustrate the underlying mechanism for the above experimental results, we further apply coarse-grained MD simulation to explore the structural properties of the polymer/IL complexes and the redistributions of different species within the complexes in response to the water and hydrophobic interfaces. The model we used is established based on our previous simulations[17,37–43] and other studies[44]. Fig. 5a shows the schematics of the coarse-grained model for different species, including the random block copolymer composed of BzMA and HEMA, IL, water, crosslinker and the bead of the hydrophobic PET substrate. Detailed illustrations of the model and the simulations are shown in the Supplementary Methods. The simulations are performed by utilizing the Large-scale Atomic/Molecular Massively Parallel Simulator (LAMMPS) software package[45].

We first explore the detailed structures of the bulk IL/polymer complexes at as-prepared and hydrated states (with 4 wt% water uptake). Figure 5b and c display the corresponding simulation snapshots, respectively. Herein, we calculate the static structure factor $S(q)$ and vertical distributions $d_v(r)$ of different species, as well as radial distribution functions $g(r)$ between different species in the polymer/IL complexes. Our simulations indicate that due to their hydrophobicity, the ILs can disperse in the random copolymers, which results in the uniform distribution of the as-prepared IL/polymer complex, as shown by the snapshot, $S_{BzMA}(q)$, $d_v(r)$, $S_{HEMA/IL/water}(q)$ (Fig. 5b, e, f) and

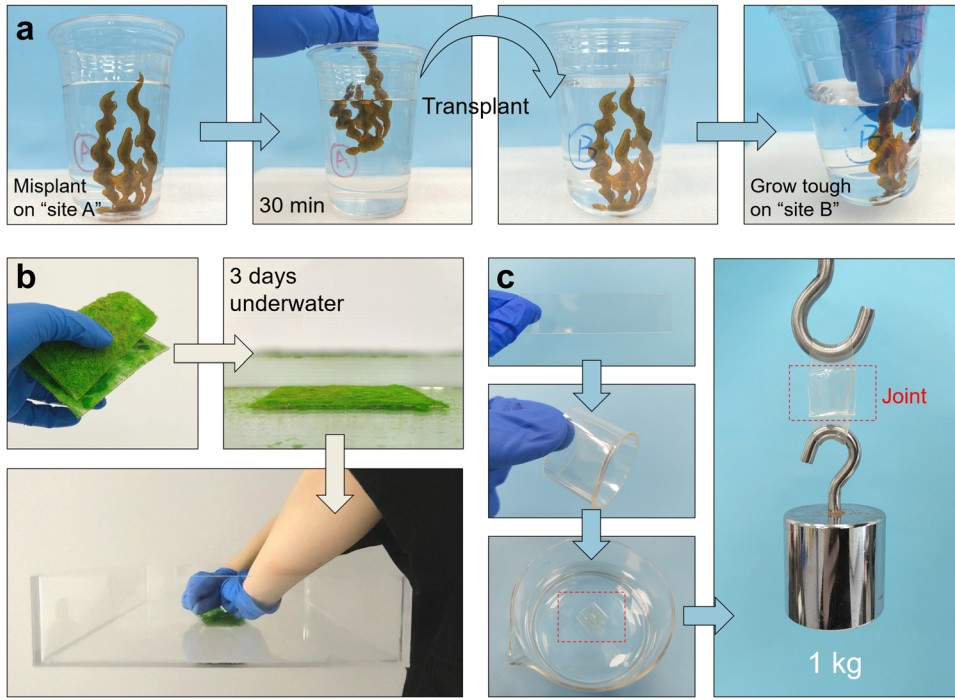

**Fig. 6 | Demonstrations of AUA. a** A design of transplantable artificial alga fabricated by AUA. The AUA alga was misplanted on "site A" and then transplanted to "site B" after 30 minutes, demonstrating its autonomous adhesive property in artificial seawater. **b** Photograph of a patch of AUA (decorated by plastic grasses) holding a PMMA tank with a bonding area of 40 cm². **c** A demonstration of the self-fusing property of AUA tape by assembling a simple lapped ring.

$g_{EMI-BzMA}(r)$ (Supplementary Fig. 20a, b, c). In the hydrated polymer/IL complex, the water molecules grasp the HEMA and ILs through hydrophilic and solvation interactions[38,42,46], whereas the BzMA aggregate through hydrophobic interactions, which causes the formation of nanoscale hydrophilic and hydrophobic domains, as shown by the snapshot (Fig. 5c), and variations in $S_{BzMA}(q)$, $d_v(r)$, $S_{HEMA/IL/water}(q)$ and $g_{EMI-BzMA}(r)$ as well as the significant peak in $g_{EMI-water}(r)$ (Supplementary Fig. 20d).

Moreover, the representative simulation snapshots of the hydrated polymer/IL complex between two water phases and between a water phase and a hydrophobic substrate is displayed with the corresponding $d_v(r)$ (Supplementary Fig. 21). As indicated by the simulations, for polymer/IL complex between two water phases, the hydrophilic HEMA components enrich at the interfaces between the complex and water phases, as shown by the peaks of HEMA at $r = 0$ nm and $r = 8$ nm in $d_v(r)$ (Supplementary Fig. 21b). Note that due to the chain connectivity, the conjunctive BzMA beads in the random copolymers are dragged close to the interfaces, as shown by the peaks of BzMA at $r = 0.6$ nm and $r = 7.4$ nm in $d_v(r)$, verifying the rationality of the result in Fig. 4. Meanwhile, the ILs are repelled inward the complex and solvated by the water molecules inner the complex (Supplementary Fig. 21c). For the cases of polymer/IL complex between a water phase and a hydrophobic substrate, we find the enrichment of HEMA near the water phase due to the hydrophilic interactions and the aggregation of BzMA on the PET substrate through hydrophobic interactions, as shown by the simulation snapshot, the water-adhesive interface and substrate-adhesive interface of the polymer/IL complex (Fig. 5d), and the peak of HEMA at $r = 8$ nm and the peak of BzMA at $r = 0$ (Fig. 5g). In addition, we find the apparent secondary enrichments of BzMA and HEMA at $r = 0.6$ nm and $r = 7.4$ nm due to the polymer chain connectivity, as well as the redistribution of IL and inner water within the complex (Supplementary Fig. 21d).

Our simulations thus expound the molecular mechanism for structural variations of polymer/IL complexes caused by hydration and rearrangement of the components due to the adhesions to the water phase and hydrophobic PET substrate. We thereby can confirm that complexion of BzMA/HEMA random copolymers and [EMI⁺/TFSI⁻] ILs can serve as an effective strategy to fabricate such AUA material with the ideal intrinsic structural self-organization and amphiphilic response capacities.

## Demonstrations

As a demonstration, AUA was employed to fabricate an artificial alga. The as-prepared "alga" was attached to the inwall of a plastic cup (marked 'A') filled with artificial seawater. After 30 minutes, the "alga" was easy to be removed from the wall, and we further transplanted it to another plastic cup (marked 'B') filled with artificial seawater. After 24 hours, the "alga" clung to the wall and became hard to remove (Fig. 6a). Such a transplantable material with gradually established adhesion was put forward in wound closure for surgeries[47], which usually involves complex chemical pretreatments and formation of covalent bonds, whereas our work provides a facile and economical method to realize this function in underwater engineering scenarios.

Furthermore, to exhibit the bearing capability of AUA, we fabricate a patch of AUA (decorated by plastic grasses) and then let it grow on the bottom of a filled PMMA water tank for 3 days. With a bonding area of 40 cm², the grown patch could sustain around 2 kg (weight of the tank) by holding the peeled edge without detaching (Fig. 6b). The self-fusing property was shown by curing an AUA tape ring underwater with a 3 × 3 cm² joint, which can easily hold a weight of 1 kg (Fig. 6c). Additionally, the adhesion tests under 3 types of simulative natural conditions (knife scraping, oil painting and mud contaminating) were performed to mimic the diversity and complexity of the natural environment (Supplementary Fig. 22). The results shows that our AUA material can overcome such harsh conditions in some extent. These demonstrations vividly displayed the unique transplantable property, excellent self-adhesion, considerable bulk strength and environmental adaptability in practical application.

## Discussion

In summary, we have successfully developed a universal yet robust strategy for the design and fabrication of an autonomous underwater adhesive. Instead of treating water molecules as an unfavorable factor, our strategy takes advantage of the ubiquitous water as a beneficial factor for the growth of the adhesive interface, as driven by water-induced rearrangement of the functional groups of the AUA material and its strong yet non-erodible interactions with the interface. The AUA material composed of an amphiphilic polymer matrix swollen with hydrophobic IL showed little to none instantaneous underwater adhesion onto the PET substrate, whereas its adhesion energy on the substrate can grow more than 5 times to 458 J·m$^{-2}$ after 24 hours. More importantly, it can adhere autonomously on most common substrates (especially on plastics) underwater without external pressing force. Comprehensive studies on macroscopic mechanical property, plasticization effect and surface hydrophobicity, as well as MD simulations have been conducted to gain a fundamental understanding on this unique adhesion process. Given the universality in material design, easy preparation, and superior underwater adhesion capacity that AUA materials have, this material can be applied in complex, harsh and changeable underwater application scenarios to help ocean exploration, as well as promising yet challenging applications in wet adhesions. Futhermore, AUA brings enlightens on developing the spontaneous behavior of materials, promoting a back-to-nature lifestyle. We expect that the minimisation of human intervention, passive energy input and the usage of additional equipment will evolve into a global trend in engineering scenarios.

## Methods

### Materials

Benzyl methacrylate (BzMA, 98%, Energy Chemical), ethyl methacrylate (EMA, 99%, Aladdin), 2-hydroxyethyl methacrylate (HEMA, 99%, Aladdin), dimethylacrylamide (DMAA, 98%, Energy Chemical), 1-ethyl-3-methyl imidazolium bis(trifluorosufonylmethane imide) (EMITFSI, 99%, Lanzhou Greenchem), 1-butyl-3-methyl imidazolium hexafluorophosphate (BMIPF$_6$, 99%, Lanzhou Greenchem), divinylbenzene (DVB, 80%, Energy Chemical), 1-hydroxycyclohexyl phenyl ketone (PI-184, 98%, Aladdin), N,N-dimethylformamide (DMF, 99.9%, Energy Chemical), tetrahydrofuran (THF, 99.9%, Energy Chemical), sodium chloride (NaCl, 99.9%, Energy Chemical), calcium chloride (CaCl$_2$, 97%, Sigma-Aldrich), magnesium chloride (MgCl$_2$, 99%, Macklin), potassium chloride (KCl, 99.5%, Energy Chemical), sodium bicarbonate (NaHCO$_3$, 99.5%, SCR) and magnesium sulfate (MgSO$_4$, 99.9%, Energy Chemical) were used as received.

### Synthesis of AUA tapes

All the AUA tapes were prepared through free radical polymerization. BzMA, HEMA, EMITFSI, DVB, and photo-initiator 184 were mixed to form a homogeneous precursor solution. The precursor solution was then degassed and injected into a clean glass mold separated by two release films and a PDMS spacer. The thickness of the tapes was fixed at 1 mm. The polymerization was carried out under UV light for 1 hour. After polymerization, the tapes were placed in an 80 °C oven for 12 hours to remove the residual monomers (Supplementary Fig. 23) and then stored in a desiccator.

### Immersion treatment

The as-prepared AUA was attached to the substrates underwater and slightly pressed with a small preload for 10 s, noted that no further loading and supplementary techniques were applied during the immersion since the tape has been proved to be pressure insensitive (Fig. 2d). Without specified otherwise, the immersion was carried out in DI water. Artificial seawater was prepared by dissolving NaCl, CaCl$_2$, MgCl$_2$, KCl, MgSO$_4$ and NaHCO$_3$ in DI water to form a homogeneous solution following the recipe in Supplementary Table 3.

### Adhesion test

Adhesion tests were performed using a tension machine (UTM 6104, SUNS Technology). For the 90-degree peeling test, samples were cut into rectangular strips of 5-10 mm in width, and 60-80 mm in length. A non-stretchable tape (FLEX TAPE) was employed as a backing to prevent the elongation of the AUA during the test. Before starting the test, partial of the sample was carefully peeled and fixed on the upper clamp. Then, the peeling tests were performed at a constant peeling speed of 100 mm/min. The adhesion energy, $\Gamma$, was determined by $\Gamma = F_{plateau}/w$, where $F_{plateau}$ is the measured force at the plateau segment, and $w$ is the width of the sample.

## Data availability

The data generated in this study including methods, tables, figures and datasets are provided in the Supplementary Information/Source Data file. Source data are provided with this paper. Additional data are available from the corresponding author upon request. Source data are provided with this paper.

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

## Acknowledgements

This work was supported by the Program for Guangdong Introducing Innovative and Entrepreneurial Teams (Grant No. 2017ZT07C291) to S.Z., the National Natural Science Foundation of China (Grant No. 22078276, 22005260 and 22205190) to Q.Z., H.Z. and X.M., the Shenzhen Science and Technology Program (Grant No. KQTD20170810141424366 and RCBS20210609103645021) to S.Z. and X.M., the Shenzhen Key Laboratory of Advanced Materials Product Engineering (Grant No. ZDSYS20190911164401990) to S.Z. Q.Z thanks the Shenzhen Stability Science Program and the Presidential Fund (Grant No. PF01000949) for supporting his research at CUHK-Shenzhen. The MD simulations of this work are financially supported by the National Natural Science Foundation of China (Grant No. 22073094), the Science and Technology Development Program of Jilin Province (Grant No. 20210402059GH), the Science and Technology Plan Projects of Yunnan Province (Grant No. 202101BC070001-007) and the Major Science and Technology Projects for Independent Innovation of China FAW Group Co., Ltd (Grant No. 20220301018GX) to X.D. We are grateful for the essential support of the Network and Computing Center, CIAC, CAS, and the Computing Center of Jilin Province.

## Author contributions

L.Y. and Q.Z. conceived the idea and designed the research. L.Y., X.M. and Z.C. carried out the material synthesis, adhesion test, static/dynamic mechanical test, surface characterization, spectroscopic analysis and demonstration design. L.Y. and Q.Z. analysed and interpreted the experimental results. C.L. and X.D. performed the computational simulation and analyased the results. L.Y. and Q.Z. drafted the manuscript, and H.Z., S.Z commented and revised the manuscript. Q.Z. supervised the project, and all authors contributed to the writing of the manuscript.

## Competing interests

The authors declare no competing interests.
