## [Peer Review File · Nature Communications]

Autonomous underwater adhesion driven by water-induced interfacial rearrangementREVIEWER COMMENTS

Reviewer #1 (Remarks to the Author):

Developing adhesives that can be used underwater has gained considerable interest, yet still remains very challenging to meet the complicated application scenarios. In this work, the authors developed an unprecedented autonomous underwater adhesion (AUA) material whose adhesion energy would increase during water immersion. Instead of treating water molecules as an unfavorable factor, the authors take advantage of the ubiquitous water as a beneficial factor for the growth of the adhesive interface. This new concept has not been reported previous, and should be of interest to the readership in the related areas.

Meanwhile, the strategy developed here is practical, the manuscript is well presented, and the AUA working mechanism has been studied comprehensively via both experimental characterizations and computational calculations. I therefore would recommend the publication of this work after the authors consider the following issues.

1. What's the contribution of ionic liquid in the AUA material design? What would happen if ionic liquid is removed? Do the authors have such results for reference? Would ionic liquid in the polymeric network diffuse into the aqueous environment?
2. After water immersion, are there any observable hydrophobic or hydrophilic domains formed in this AUA material?
3. It will be more intuitive to use dark red in Figure 4g to represent BzMA or benzyl group in line with Figure 4d and 4f.
4. Water molecule is the key driving force of such autonomous adhesion. Would it work if first immersing the AUA tape in water for hours and then adhering it to the substrate?

Reviewer #2 (Remarks to the Author):

The manuscript by Duan et al describes a new formulation of underwater adhesive tape that is made by a random copolymerization of Bz-MA and HEMA in ionomer (IL). This formulation possesses sufficient fluidity and interfacial amphipathy to enable synergistic rearrangements that result in the time-dependent i) absorption of surface water and ii) coupling of nonpolar Bz functionalities with nonpolar surfaces. Both these tendencies are convincingly

demonstrated on PET, however, other results and interpretations are less convincing.

Points of critical comment:

1. P. 5: The adhesive's best Eadh performance is limited to nonpolar polymer surfaces (not including PE and PP). How likely are these to be significant target surfaces?
2. In Fig 3e & f, what substratum was tested?
3. Neither modeling nor experiments provide satisfactory explanations for why observed Eadh on PE and PP surfaces wasn't high. As PE and PP are among the most nonpolar surfaces tested, does it not cast doubt on hydrophobicity as the driver? Instead, specific enthalpic interactions may play a role?
4. In figure 5, models in d of interfacial molecular redistributions on a PET surface show close juxtaposition of Im and Bz. To what extent is this based on Cation-pi interactions?
5. All surfaces exposed to natural aqueous media (physiological fluids, rivers, seawater etc) become effectively fouled by films of conditioning and microbial films over time. How would AUA tapes interact with these films?
6. We live in a new age of awareness in which sustainability is seen as more important to survival than convenience. How does this invention enhance sustainability?

Reviewer #3 (Remarks to the Author):

The paper entitled "Autonomous underwater adhesion driven by water-induced interfacial rearrangement" by Le Yao, Chengjiang Lin, Xiaozheng Duan, Xiaoqing Ming, Zhixuan Chen, He Zhu, Shiping Zhu and Qi Zhang, presents an interesting work on a polymerized network confining ionic liquid. The observed property (underwater adhesion) is worth to be published. The conclusions are drawn from experimental evidences. Few details could be upgraded (like in the sentence "hydrophilic component, i.e., bis(trifluorosulfonylmethane imide) lithium salt (LiTFSI)" please precise whether Li or TFSI ; or like in the paragraph "Meanwhile, the adhesion performance of AUA tapes was evaluated on various substrates, including polyethylene (PE), polypropylene (PP), polyvinyl chloride (PVC), polystyrene (PS), polymethyl methacrylate (PMMA), glass and steel (Figure 2f).", where no conclusion is given – except "more significant on plastic substrates" -). Nevertheless, the main concern relies in

the novelty although the authors are using the term “unique” quite often : monomers cured in the presence of ionic liquids have already been published, for diverse application, and no reference to these work are provided here : e.g. C. Gerbaldi et al. J. Membr. Sci. 459 (2012) 423–424, D. Aidoud et al. J. Electrochem. Soc. 165 (13) A3179-A3185 (2018), M.D. Dickey et al. Nature Mater. VOL 21 March 2022, pp359–365 .

This although the work deserve publication after upgrading the referencing, it seems that it does not reach the standards for publication by Nature group.

Reviewer #1 (Remarks to the Author):

Developing adhesives that can be used underwater has gained considerable interest, yet still remains very challenging to meet the complicated application scenarios. In this work, the authors developed an unprecedented autonomous underwater adhesion (AUA) material whose adhesion energy would increase during water immersion. Instead of treating water molecules as an unfavorable factor, the authors take advantage of the ubiquitous water as a beneficial factor for the growth of the adhesive interface. This new concept has not been reported previous, and should be of interest to the readership in the related areas. Meanwhile, the strategy developed here is practical, the manuscript is well presented, and the AUA working mechanism has been studied comprehensively via both experimental characterizations and computational calculations. I therefore would recommend the publication of this work after the authors consider the following issues.

R: Thanks very much for the positive comments on the novelty of our work. We sincerely appreciate your time and effort in providing valuable feedback on our manuscript. Please kindly find our point-to-point reply below.

Q: What's the contribution of ionic liquid in the AUA material design?

R: Thanks for your insightful comments. As compared with polymer elastomer adhesives without ionic liquid, ionogel adhesives can dissipate more energy during the deformation, attributed to their high viscosity and abundant interionic interactions. Current works have proven that the adhesion performance could be improved by introducing ionic liquids into adhesives (*Adv. Mater.* 2021, 33, 2103174; *Adv. Funct. Mater.* 2021, 31, 2009334). On the other hand, there is a subtle dissolution equilibrium between this ionic liquid and some of the methacrylate polymers, and the related discussions are shown in the main text (Page 5 highlighted): “As a typical hydrophilic polymer, poly(2-hydroxyethyl methacrylate) (PHEMA) is not compatible with most hydrophobic ILs, such as 1-ethyl-3-methyl imidazolium bis(trifluorosulfonylmethane imide) (EMITFSI). Surprisingly, the random copolymers of BzMA and HEMA show good miscibility with EMITFSI, indicating a subtle balance of interactions between the polymer and IL (Supplementary Fig. 1)”. Therefore, this IL-based system was chosen for the design of AUA material.

Q: What would happen if ionic liquid is removed? Do the authors have such results for reference?

R: Thanks for your valuable comments. Yes, we have prepared additional samples without ionic liquid, and the corresponding stress-strain curves have been added in Supplementary Fig. 13. It's been found that when the ionic liquid was removed, the material became stiff and brittle plastic, showing little to none adhesion.

Revised: Photographs of the sample at $I = 0$ and the corresponding stress-strain curves have been added in Supplementary Fig. 13. Related discussions have been supplemented as in the revised manuscript (Page 12 highlighted): “At $I = 0$, the material will turn into a brittle plastic without any adhesion (Supplementary Fig. 13a). Obviously, EMIFTSI not only provides a subtle dissolution equilibrium and interionic interactions to the AUA process, but also plays an important role in regulating the bulk mechanical properties (Supplementary Fig. 13b).”

Supplementary Fig. 13 a Photographs showing the rigidity and brittleness when the ionic liquid was removed. **b** Stress-strain curves of AUA at different I .

Q: Would ionic liquid in the polymeric network diffuse into the aqueous environment?

R: Thanks for your constructive comments. During the long-term immersion, it is inevitable that there would be some ionic liquid diffusing into the water, but the amount is very tiny. We measured the weight loss of the AUA materials by weighing the dry samples after different immersion time.

Revised: The measured results have been added in Supplementary Fig. 11 and the related discussions have been supplemented as “Owing to the covalently crosslinked network, there will be only a tiny amount of ionic liquid loss (2.85 wt% after 7 days), which is negligible compared to the total ionic liquid content (40 wt%) (Supplementary Fig. 11)” in the main text (Page 11 highlighted).

Supplementary Fig. 11 Weight loss of the bulk AUA material immersion recorded by weighing at different immersion time. The tested samples were obtained after being immersed in DI water for 1-7 days and then dried in a 110 °C oven until reached weight equilibrium.

Q: After water immersion, are there any observable hydrophobic or hydrophilic domains formed in this AUA material?

R: Thanks for your valuable comments. There were no observable domains formed in the AUA material. The distribution of the BzMA and HEMA units on polymer chains is considered to be random since our AUA materials were prepared through conventional free radical polymerization. Therefore, the effects of polymer chain connectivity can confine the hydrophobic units and hydrophilic units, which prevents the occurrence of phase separation. As a supplement, we further analysed the microphase separation by small-angle X-ray scattering (SAXS). The result shows no scattering peak at both $t = 0$ and $t = 24$ h.

Revised: The SAXS profiles have been added in Supplementary Fig. 19 and the related discussions have been supplemented as “Interestingly, the competition between the hydrophobicity units (BzMA) and hydrophilic units (HEMA) did not evolve into

observable phase separation. The small-angle X-ray scattering (SAXS) profiles confirmed this statement that neither the pristine sample nor the 24-hour-immersed sample showed any scattering peak (Supplementary Fig. 19). This might be attributed to the randomness of the free radical polymerization and the conformal constrain of the crosslinked polymer chain. Thus, we attempted to use computational methods to elucidate the more microscopic process in the following chapter” in the main text (Page 15, 16 highlighted). The corresponding method has also been supplemented (Page 24 highlighted).

Supplementary Fig. 19 SAXS profiles of AUA comparing t = 0 and t = 24 h.

Q: It will be more intuitive to use dark red in Figure 4g to represent BzMA or benzyl group in line with Figure 4d and 4f.

R: Agreed and thanks for the suggestion. The color of the BzMA unit has been modified in Fig. 4g (Page 13 highlighted).

Fig. 4 g An illustration of the rearrangement of the ester, benzene and hydroxyl groups when introduced with water. (AUA tapes at $f = 0.5$, $I = 40\%$, and $X = 0.15\%$ were chosen throughout the test.)

Q: Water molecule is the key driving force of such autonomous adhesion. Would it work if first immersing the AUA tape in water for hours and then adhering it to the substrate?

R: Thanks for your insightful comments. If first immersing the AUA tape in water for hours and then adhering it to the substrate, it will not form any interfacial adhesion.

Revised: The related discussions have been supplemented as “Besides, the upper surface is fully non-adhesive, indicating that the autonomous adhesion is only formed at specific interfaces” in the main text (Page 14 highlighted).

Reviewer #2 (Remarks to the Author):

The manuscript by Duan et al describes a new formulation of underwater adhesive tape that is made by a random copolymerization of Bz-MA and HEMA in ionomer (IL). This formulation possesses sufficient fluidity and interfacial amphipathy to enable synergistic rearrangements that result in the time-dependent i) absorption of surface water and ii) coupling of nonpolar Bz functionalities with nonpolar surfaces. Both these tendencies are convincingly demonstrated on PET, however, other results and interpretations are less convincing.

R: We sincerely appreciate your time in providing valuable and insightful comments on our work. Please kindly find our point-to-point reply in the following.

Points of critical comment:

Q: P. 5: The adhesive's best E_{adh} performance is limited to nonpolar polymer surfaces (not including PE and PP). How likely are these to be significant target surfaces?

R: Thanks for your valuable comments. Actually, the AUA material does not only target to nonpolar surfaces, but also showed growing adhesion on substrates with certain polarity such as PVC (Fig. 2f), which is one of the main materials to build water pipelines. Overall, these polymer substrates (PET, PS, PMMA and PVC) are widely used and they are likely to be significant target surfaces of AUA.

Q: In Fig 3e & f, what substratum was tested?

R: Thanks for the careful reading of our manuscript. In Fig 3e & f, the 90-degree peeling tests were performed on PET substrates. We have added this information accordingly in the manuscript (Page 9 highlighted).

Q: Neither modeling nor experiments provide satisfactory explanations for why observed E_{adh} on PE and PP surfaces wasn't high. As PE and PP are among the most nonpolar

surfaces tested, does it not cast doubt on hydrophobicity as the driver? Instead, specific enthalpic interactions may play a role?

R: Thanks for your valuable comments. Yes, “PE and PP are among the most nonpolar surfaces tested”, while they are also two of the most typical low surface energy (LSE) materials with exceptional non-wetting properties, making them extremely hard to adhere without surface modification. It’s been found that most of the previously reported adhesives had lower adhesion strength on LSE substrates among the others (*ACS Appl. Mater. Interfaces* 2021, 13, 41112–41119; *Euro. Polym. J.* 2020, 137, 109949; *Adv. Mater.* 2012, 24, 5676-5680). Therefore, it is as expected that our AUA tape showed relatively lower adhesion energy on PE and PP surfaces.

Although the adhesion energy on PE and PP after water immersion for 24 h is not numerically high, the contrast (Γ_{24h}/Γ_{0h}) can still reach a similar level as other polymer substrates, i.e., 4.3 and 4.5 times for PE and PP, respectively. Notably, all the other commonly used plastic substrates showed 4-6 folds of contrast, whereas, for other hydrophilic substrates such as glass and steel, Γ showed less growth after 24-hour immersion. We also have conducted additional water contact angle tests on all the tested substrates (Supplementary Fig. 17). The results show that all plastic substrates exhibit contact angles between 80° and 105°, indicating considerable hydrophobicity. In contrast, the hydrophilic glass and steel substrates with the most indistinctive contrast showed low water contact angles of 21° and 51°, respectively. This clearly indicates that surface hydrophobicity plays a dominant role in the formation of strong adhesion.

Some revisions have been made accordingly (Page 14 highlighted).

Supplementary Fig. 17 Contact angle of water on the tested substrates.

Q: In figure 5, models in d of interfacial molecular redistributions on a PET surface show close juxtaposition of Im and Bz. To what extent is this based on Cation- π interactions?

R: We thank the reviewer for the insightful comment. The ILs used in our study include both the ionic groups and partial hydrophobic groups, and therefore, in our simulation, we coarse-grain the ILs as amphiphilic beads according to the classical MARTINI force field (*J. Phys. Chem. B* 2007, 111, 7812-7824). In this context, both the polymer (BzMA groups) and the ILs show attractive interactions with the hydrophobic PET substrate. In addition, we also consider the comprehensive affinity between ILs and polymers caused by both the weak hydrophobic attraction and Cation- π attractions using the Lenard-Jones potential according to the aforementioned classical MARTINI force field. All these comprehensive effects cooperate to result in the juxtaposition of ILs and BzMA beads on the PET substrate. If the Cation- π interaction is fully ignored, the similar juxtaposition phenomenon of ILs and BzMA beads can be reserved because of the hydrophobic attraction between IL and PET.

Q: All surfaces exposed to natural aqueous media (physiological fluids, rivers, seawater etc) become effectively fouled by films of conditioning and microbial films over time. How would AUA tapes interact with these films?

R: Agreed and thanks for your insightful suggestion. To investigate the universality and robustness of our AUA tapes, we have further conducted peeling tests under three simulative natural conditions including the underwater scraped surface, underwater oil-painted surface and muddy aqueous environment, to mimic the diversity and complexity of the natural environment. To our delight, considerable AUA performance has been observed under all these three testing conditions, indicating the robustness of our AUA tapes (Supplementary Fig. 22).

Revised: The results have been added in Supplementary Fig. 22 and the related discussions have been supplemented as “Additionally, the adhesion tests under 3 types of simulative natural conditions (knife scraping, oil painting and mud contaminating) were performed to mimic the diversity and complexity of the natural environment (Supplementary Fig. 22). The results shows that our AUA material can overcome such harsh conditions in some extent. These demonstrations vividly displayed the unique transplantable property, excellent self-adhesion, considerable bulk strength and environmental adaptability in practical application” in the main text (Page 20 highlighted).

Supplementary Fig. 22 a Photographs of three 3 types of simulative natural conditions: **i** underwater scraped surface, **ii** underwater oil painted surface and **iii** muddy aqueous environment, respectively (substrate: PET). **b** The corresponding force-displacement curves of AUA during 90-degree peeling on these conditions.

Q: We live in a new age of awareness in which sustainability is seen as more important to survival than convenience. How does this invention enhance sustainability?

R: Thanks for your valuable comments. In the new era, sustainability is a common goal of all mankind. Reducing energy consumption is one of the major principles of sustainability. The concept of AUA mainly focuses on the spontaneity of adhesive materials, which is intended to minimize the passive energy input and the usage of additional equipment during the underwater adhesion operation. Moreover, such spontaneous behavior of materials largely replaces human intervention in engineering scenarios, promoting the development of a back-to-nature lifestyle.

Revised: Accordingly, we revised the discussion as “Given the universality in material design, easy preparation, and superior underwater adhesion capacity that AUA materials have, this conceptually novel material can be applied in complex, harsh and changeable underwater application scenarios to help ocean exploration, as well as promising yet challenging applications in wet adhesions. Futhermore, AUA brings enlightens on developing the spontaneous behavior of materials, promoting a back-to-nature lifestyle. We expect that the minimisation of human intervention, passive energy input and the usage of additional equipment will evolve into a global trend in engineering scenarios” in the main text (Page 21 highlighted).

Reviewer #3 (Remarks to the Author):

The paper entitled “Autonomous underwater adhesion driven by water-induced interfacial rearrangement” by Le Yao, Chengjiang Lin, Xiaozheng Duan, Xiaoqing Ming, Zhixuan Chen, He Zhu, Shiping Zhu and Qi Zhang, presents an interesting work on a polymerized network confining ionic liquid. The observed property (underwater adhesion) is worth to be published. The conclusions are drawn from experimental evidences.

R: We are very grateful to the Reviewer for the valuable comments on our manuscript. We have revised the manuscript and hope to address the issues that you raised.

Few details could be upgraded (like in the sentence “hydrophilic component, i.e., bis(trifluorosulfonylmethane imide) lithium salt (LiTFSI)” please precise whether Li or TFSI;

R: Thanks for your careful reading of our manuscript. In bis(trifluorosulfonylmethane imide) lithium salt, the lithium ion is the main hydrophilic part, which has been extensively discussed in the previous works (*Aggregate* 2023, 4, e249; *Chem. Commun.* 2015, 51, 5448; *Macromolecules* 2017, 50, 4780). Revision has been made accordingly (Page 4 highlighted).

Or like in the paragraph “Meanwhile, the adhesion performance of AUA tapes was evaluated on various substrates, including polyethylene (PE), polypropylene (PP), polyvinyl chloride (PVC), polystyrene (PS), polymethyl methacrylate (PMMA), glass and steel (Figure 2f).”, where no conclusion is given – except “more significant on plastic substrates” -).

R: Thanks for your valuable suggestion. To better understand the relation between the AUA performance and the substrate properties, we supplemented the contact angle analysis on different substrates in Supplementary Fig. 17 and further conclusions have been made.

Revised: The contact angles on different substrates have been presented in Supplementary Fig. 17 and the related discussions have been rewritten as “As the reference of

hydrophobicity, we measured the contact angle of water on all the tested substrates. The water contact angle on the plastic substrates was between 80° to 105°, which is close to the contact angle of AUA, while the glass and steel substrates are more hydrophilic (< 55°) (Supplementary Fig. 17). This result could be a rational explanation for why AUA is more significant on plastic substrates. Moreover, such obvious contrast between the two water-plasticized surfaces points to a directional motion of the functional groups during the interfacial adhesion development” in the main text (Page 14 highlighted).

Supplementary Fig. 17 Contact angle of water on the tested substrates.

Nevertheless, the main concern relies in the novelty although the authors are using the term “unique” quite often: monomers cured in the presence of ionic liquids have already been published, for diverse application, and no reference to these work are provided here: e.g. C. Gerbaldi et al. J. Membr. Sci. 459 (2012) 423–424, D. Aidoud et al. J. Electrochem. Soc. 165 (13) A3179-A3185 (2018), M.D. Dickey et al. Nature Mater. VOL 21 March 2022, pp359–365.

This although the work deserve publication after upgrading the referencing, it seems that it does not reach the standards for publication by Nature group.

R: We sincerely appreciate the valuable comments from the reviewer, yet we respectfully disagree with the comment of “the main concern relies in the novelty”. We have carefully read the suggested reference, which reported the preparation of ionogels through *in situ*

polymerization, and their applications in various fields, such as lithium batteries, and wearable devices. We agree with the reviewer that the preparation of ionogels is not new, and ionogel adhesive has also been reported. Nevertheless, these are NOT our key points reported in the current work. To the best of our knowledge, our material in this work has achieved the first case of showing strong and spontaneous adhesion underwater. As Reviewer #1 commented, “This new concept has not been reported previously, and should be of interest to the readership in the related areas”. To help the reviewer better understand the novelty of our work, here are some points to highlight:

(1) We developed the first autonomous underwater adhesion material, which had little to none underwater adhesion at the beginning, whereas the adhesion energy on the substrate can grow strongly with prolonged time.

(2) Instead of treating water molecules as an unfavorable factor, the strategy we developed in this work takes advantage of the ubiquitous water as a beneficial factor for the growth of the adhesive interface.

(3) Through both experimental characterizations and computational calculations, it has been confirmed that this unique underwater adhesion process is driven by water-induced rearrangement of the functional groups.

Therefore, we still believe that our work, which focuses on the conceptually new AUA materials, can meet the novelty, importance and urgency criteria of *Nat. Commun.* and could be suitable for publication in this journal.

Revised: The references mentioned by the reviewer have been cited in the main text (Page 5 highlighted).

REVIEWERS' COMMENTS

Reviewer #1 (Remarks to the Author):

I would like to suggest the acceptance of this revised manuscript.

Reviewer #2 (Remarks to the Author):

Reviewer thanks authors for their thoughtful responses and additional experiments using "field-relevant" aqueous test solutions.

Reviewer #3 (Remarks to the Author):

the answers and discussion are valid; the modifications made are relevant. The paper could be published.